

# Effects of wheat stubble on runoff, infiltration, and erosion of farmland in the Loess Plateau, China subjected to simulated rainfall

Linhua Wang[a], Bo Ma[a], and Faqi Wu[a,b] [*]
[a] Institute of Soil and Water Conservation, Northwest A&F University, Yangling 712100, PR China
[b] College of Natural Resources and Environment, Northwest A&F University, Yangling 712100, PR China
[*] Correspondence to: Faqi Wu (wufaqi@263.net)

**Abstract**

Soil and water losses in agriculture are major environmental problems worldwide, especially on
the Loess Plateau, China. Summer fallow management may help to control soil erosion and conserve
water. This study investigated the effects of wheat stubble on runoff, infiltration, and soil loss in
laboratory plots under simulated rainfall. The treatments comprised wheat stubble cover (WS) and
traditional plowing (TP) in runoff plots (4.0 m ×1.0 m) with three slope gradients (5°, 10°, and 15°)
under simulated rainfall at 80 mm h$^{-1}$ for 1 h. The runoff volume from WS plots was significantly less
than that from TP. The runoff reduction with WS ranged from 91.92–92.83% compared with TP. The



runoff rates varied with the runoff volume in the same manner. Under WS, sediment losses (2.41–3.78 g

m$^{-2}$) were reduced dramatically compared with TP (304.31–731.23 g m$^{-2}$). The sediment concentration

was also significantly lower with WS than TP. The infiltration amount was higher with WS (94.8–96.2%

of rainwater infiltrated) than TP (35.4–57.1%). Thus, stubble cover can help to control erosion and

conserve soil and water resources.

**Keywords:** Loess Plateau, Soil erosion, Summer fallow, Traditional plowing, Wheat stubble,

## 1. Introduction

Soil and water losses from agricultural land, particularly sloping farmland, are regarded as major

environmental threats to ecosystem sustainability on the Loess Plateau, China. Approximately 60% of

the total watershed sediment and runoff is derived from sloped farmland due to natural and human

factors, such as the precipitation intensity, geomorphology, and soil management practices, which all

contribute to farmland degradation (Fu et al., 2000; Fu et al., 2006; Kang et al., 2001; Shi and Shao,

2000).

The susceptibility to soil and water losses is higher on farmland than land with other vegetation

types (e.g., forestry, shrub, and grass) (Boardman et al., 1990; Shi and Shao, 2000; Tang, 2004). This is

because the characteristics of crops, including the crop canopy architecture and root system, differ from

those of other vegetation types, which leads to severe soil erosion on farmland (Cerdà et al., 2009;

Gómez et al., 2004; Llorens and Domingo, 2007). In particular, crops are harvested, which means that

there is a fallow period when vegetation does not cover the soil surface. In the Loess Plateau region of



China, farmers conduct tillage practices during the summer fallow period after harvesting winter wheat
in June. The main aims of summer tillage are removing weeds, and creating a favorable rough surface to
maximize rainfall capture and minimize soil evaporation (Hammel et al., 1981; Vermang et al., 2015).
However, soil management in the summer fallow period may completely disrupt the surface soil,
thereby making farmland more susceptible to severe soil erosion in the fallow rainy season (Wang et al.,
2016).  In the Loess Plateau region, most of the annual rainfall is concentrated in the summer between
July and September, when 60–70% of the total annual rainfall occurs (Shi and Shao, 2000). Thus, the
rainfall is erratic and it occurs with a high intensity and short duration. Summer tillage of bare sloped
farmland means that extreme erosive rainfall events can cause severe surface runoff and soil erosion.
Thus, the occurrence of rainfall and the tillage of sloped farmland in the summer contribute greatly to
extreme soil erosion. This is why soil erosion from sloped farmland is recognized as the main source of
sediment losses, which are higher than those from other land use types. Therefore, it is crucial to prevent
soil erosion from sloped farmland during the summer fallow period.
To reduce soil erosion from farmland, numerous studies have considered the roles of surface
cover and soil management practices in conservation agriculture worldwide (Bescansa et al., 2006;
Jordán et al., 2010; Prosdocimi et al., 2016; Swella et al., 2015; Won et al., 2012). Retaining a surface
covered with a layer of crop residues is a suitable management practice for preventing soil losses and
conserving rainwater on farmland. Thus, Gholami et al. (2013) investigated the effects of straw mulch
on soil erosion under various rainfall intensities in laboratory simulations, where the results showed that
straw mulch was effective in delaying the runoff initiation time, as well as reducing splash erosion, the
runoff volume, and sediment losses. Kukal and Sarkar (2010) also investigated the effects of wheat



mulch on splash erosion and infiltration under simulated rainfall conditions, and concluded that straw
mulch can play an important role in mitigating splash erosion and increasing infiltration. Jordán et al.
(2010) showed that the application of wheat straw mulch could improve the physical and chemical
properties of soil, as well as reduce runoff and soil losses in cultivated land under semiarid conditions.
Prosdocimi et al. (2016) examined the effects of straw mulch on soil erodibility and surface runoff, and
found that straw cover can be effective in reducing soil erodibility, thereby decreasing the soil erosion
risk on cultivated land. Similar results were obtained by Cerdà et al. (2016), who found that straw
residues were effective in reducing soil and water losses from agricultural land under simulated rainfall
in the field. Swella et al. (2015) showed that retaining crop residues increased rainfall infiltration and
reduced evaporation during the summer and autumn. Aboudrare et al. (2006) investigated the effects of
fallow management on soil water storage during the fallow period in semiarid Mediterranean conditions,
where they found that the implementation of appropriate management could maximize the capture of
rainwater. Blanco and Lal (2008) investigated tillage practices to increase rainwater storage for winter
wheat production and runoff reduction in the dryland areas of the western USA and Great Plains during
the summer fallow periods. Lee and Yang (1965) reported that summer plowing during the summer
fallow period can improve the physical, chemical, and biological properties of soil, thereby increasing
winter wheat production in the Loess Plateau region. These studies have provided insights into the
importance of surface cover and soil management practice for controlling soil and water losses from
farmland. Beneficial strategies for soil and water conservation are based on the following principles:
providing cover to absorb raindrop kinetic energy as well as reducing splash erosion and the overland
flow velocity (Gholami et al., 2014; Gholami et al., 2013; Sadeghi et al., 2015); improving the physical



and chemical properties of soil, such as the organic matter content and stability of aggregates (Jordán et
al., 2010); increasing the soil infiltration capacity and decreasing evaporation (Adekalu et al., 2007;
Swella et al., 2015; Todd et al., 1991); and creating a rough surface to decrease runoff and enhance
infiltration (Strudley et al., 2008; Vermang et al., 2015; Wang et al., 2016). The obvious advantages of
crop residues as a soil cover in conservation agriculture are known, but the effects of crop residues on
rainwater capture and soil and water losses from sloped farmland have been studied little, especially
during the summer fallow period in the Loess Plateau region.
On the Loess Plateau, the area of cultivated land is 145 800 km$^2$, which accounts for 25.6% of
the total land area. About 70% of the cultivated land comprises rainfed areas, which are distributed in
mountainous and hilly regions (NDRC et al., 2010). On sloped farmland, a combination of continuous
and intense cultivation, inappropriate soil management, and concentrated rainfall in the summer fallow
period have caused severe soil erosion, thereby decreasing the soil productivity and increasing land
degradation. Wheat is one of the major crops in the Loess Plateau region, where it accounts about 35%
of the total cultivated area and 30% of the total crop production (NDRC et al., 2010). The availability
and costs of application with stubble cover mean that it is practical for farmers to implement this method
in the summer fallow periods. In addition, retaining stubble cover is an efficient and environmentally
friendly method that utilizes the crop biomass. The present study aimed to investigate the effects of
retaining the stubble from wheat on runoff, erosion, and rainwater capture in a laboratory plot under
simulated rainfall and different slope conditions. The results of this study provide a better understanding
of the effects of stubble cover for farmers and policy makers, which may be important for conserving
soil productivity and supporting sustainable agriculture in the Loess Plateau rainfed area.



## 2. Material and Methods

2.1 Experimental plots and rainfall simulator

This study was performed at the Laboratory of Soil and Water Conservation located at the Northwest A&F University campus, Yangling, Shannxi Province. Yangling (E107°59′–108°08′, N34°14′–34°20′) is located on the Guanzhong Plain at an altitude of 516.4–540.1 m. This area has a warm temperate, semi-humid monsoon climate with an average annual temperature of 12.9°C. The annual precipitation is 635.1 mm and the highest rainfall occurs in the period from July to September, which accounts for 60–70% of the annual precipitation. The experiments were conducted on runoff plots built in 2009, as shown in Fig. 1a. The areas of most plots used for laboratory rainfall simulations in previous studies were less than 5 m$^2$ (Huang et al., 2013; Wu et al., 2014; Zhao et al., 2013). In the present study, the runoff plot measured 4.0 m (length) × 1.0 m (width) × 0.6 m (depth), and four runoff plots comprised a slope gradient group. The slope gradients of the runoff plots were 5°, 10°, and 15°, which represented farmland with slight, gentle, and steep slopes in the field based on the classification of farmland in the Loess Plateau region, where 42% of the farmland has slopes of 5–15° (NDRC et al., 2010). The runoff plots were filled with soil, which was taken from the top 0–20 cm soil layer of a farm in Yangling after the runoff plot was constructed in July, 2009. The soil was clay loam and its major physico-chemical properties are summarized in Table 1. Before placing the soil in the plot, a 10-cm layer of sand was laid at the bottom to allow free drainage. After filling with soil, the runoff plots were left for one year to allow natural compaction in order to obtain soil properties similar to the natural



conditions. Each runoff plot had an aluminum sheet at the lower end of plot, which served as an outlet
for collecting runoff samples.
Rainfall simulation is used widely as a method for studying runoff and erosion processes. The
portable rainfall simulation system used in the present study was developed by the Institute of Soil and
Water Conservation, Chinese Academy of Science and Ministry of Water Resources, as described by
Wang et al. (2016). Its main components comprised a pumping system, inlet pipes, control valve, steel
pipes, piezometer, spray nozzles and bracket to hold the spray nozzle (Fig. 1b). Raindrops were
generated by a spray nozzle with a drop height of 7.5 m and they had comparable characteristics to
natural rainfall in terms of height. Depending on the water supplied by the pump, the rainfall intensity
varied according to different pressures displayed on the piezometer, which was installed at the inlet of
the steel pipe. The rainfall simulator system was calibrated by the pressure to obtain different intensities.
The effective cover area of the simulated rainfall was 5.0 m in length and 4.0 m in width, which was
sufficient to cover the area of the two runoff plots while avoiding border interference. Therefore,
simulated rainfall was applied simultaneously to two neighboring plots. The two rainfall simulators were
placed between neighboring runoff plots. One rainfall simulator was placed 0.5 m from the upslope plot
edge and the other was placed 0.5 m from the downslope edge. During rainfall simulations, the adjacent
plots were covered with plastic sheets to prevent rainfall falling on the soil.
2.2 Experimental treatments
The experiment was conducted in June 2013. The wheat variety used was Xiaoyan-22 and it was
sown in four plots during October 2012. Soybean (2010) and maize (2011) had been planted in the plots





for scientific research before the plots were planted with wheat. The plots were cleaned, plowed, and
prepared to obtain a seed bed, as shown in Fig. 1a. The wheat was sown in rows with a space of 20 cm
and the sowing rate was 13 g m$^{-2}$. During the wheat growth season, the crop management practices were
similar to those employed by local farmers in the field. The mature wheat was harvested in June 2013.
The two plot treatments comprised wheat stubble (WS) with a height 20 cm above the ground, whereas
the aboveground parts of the wheat biomass were cleared from other two plots and traditional plowing
was applied (TP). TP is typically used by local farmers on the sloping farmland in the Loess Plateau
region. TP uses a single plow with an iron frame and an attached blade to break up and turn the soil
across the slope direction at a depth of 20 cm.

2.3 Rainfall simulation and data analysis

After preparing all the treatments, the plot border was hydrologically isolated with plastic boards

inserted 15 cm underground and 15 cm aboveground, which prevented runoff flowing out or into
adjacent plots. A 60-min rainfall simulation at an intensity of 80 mm h$^{-1}$ was used in this experiment
based on long-term monitoring result of the natural rainfall intensity on the Loess Plateau. After starting
a rainfall simulation, the time to runoff initiation was recorded, which was determined when runoff
started to flow from the outlet of the plot. Runoff samples were collected at intervals of 2.0 min from
two plots with plastic buckets, which had been weighed previously. After finishing a rainfall simulation,
the samples were weighed and left undisturbed for 24 h. The deposited sediment was then poured into
aluminum boxes, which had been weighed previously, and the sediment was oven dried at 105°C for 24
h and then weighed again. Based on the runoff volume, sediment yield, and time interval data, the runoff



rate (mm min$^{-1}$), infiltration rate (mm min$^{-1}$), cumulative infiltration amount (mm), sediment
concentration (g L$^{-1}$), and sediment loss (g m$^{-2}$) were calculated (Zhao et al., 2014).

One-way analysis of variance was used to analyze the effects of different treatments on the

runoff rate, runoff volume, sediment concentration and loss, and the infiltration amount. Statistical
analyses were performed using IBM SPSS Statistics 19.0 (IBM, 2010). The figures were drawn using
Sigma Plot 10.0 (Systat, 2008).
**3. Results and discussion**

3.1 Runoff

Table 1 summarizes the time to runoff initiation, runoff rate, and runoff volume for different

treatment plots. Figure 2 shows the dynamics of the runoff rate during the rainfall simulations. WS
delayed the runoff initiation time by about 4–18 min compared with TP, which indicates that WS had a
positive effect on runoff generation. In general, the runoff initiation time decrease in the order of WS >
TP with the three slope gradients, and there was a significant difference between the two treatments ($P <$
0.05). In addition, the runoff initiation time decreased as the slope increased, as shown by Yair and
Lavee (1976). However, the difference in the runoff initiation time between WS and TP decreased as the
slope increased, although the difference was not significant. For example, the difference was 18.9 min at
5° and 4.06 min in 15°. Thus, WS cover had limited effects on delaying the initiation of runoff with
relatively steep slopes.

WS had a lower runoff rate and runoff volume, as shown in Table 1. There were significant

differences between WS and TP in terms of both the runoff rate and runoff volume ($P < 0.05$). The



runoff volumes were 37.86, 47.28, and 51.42 mm under TP with the three slopes of 5°, 10°, and 15°,
respectively, but only 3.00, 3.39, and 4.16 mm under WS. The runoff volume under WS was reduced by
91.92–92.83% compared with that under TP. The runoff rate varied in the same manner as the runoff
volume. The dynamics of the runoff rate were also captured, and the runoff processes varied greatly
among treatments. In general, the runoff rate response seemed to be less sensitive to rainfall, where there
was a slight increasing trend in the initial runoff stages, before remaining at low values during the
rainfall simulation under WS compared with TP. Under TP, the runoff rate increased during the first 25–
30 min, before stabilizing after 40 min with all three slopes. These results demonstrate that WS direct
affected the delay in runoff generation and reduced the amount of runoff. These findings are similar to
the results reported by Jordán et al. (2010), who found that mulch cover significantly reduced the runoff
compared with bare slope. Similarly, Puustinen et al. (2005) found that the presence of mulch facilitated
infiltration and delayed runoff, thereby resulting in much less runoff. In addition, Won et al. (2012)
found that straw mat cover resulted in significant less runoff because the residues acted as a barrier to
reduce the overland flow velocity, thereby allowing more rainfall to infiltrate, which minimized the
runoff rate and reduced the runoff volume. Therefore, these results demonstrate that WS can
significantly reduce runoff.

3.2 Soil infiltration

The soil water content is a limiting factor that affects crop yields in rainfed farmland in semiarid

regions, especially on the Loess Plateau. Tillage during summer fallow in the Loess Plateau region aims
to retain rainwater and conserve soil water moisture for the next season's crop. Precipitation is the main
source of soil moisture for agronomic crops in this area and it measures the proportion of rainwater that



infiltrates into the soil, thereby providing insights into the effects of WS cover on soil water
conservation.

Figure 3 shows the dynamic infiltration processes for all the treatments with various slope

gradients under simulated rainfall, which indicates that there was a higher initial infiltration rate under
TP, but it decreased rapidly as the simulated rainfall proceeded and reached a steady infiltration rate with
small fluctuations at 25–30 min, as observed by Wang et al. (2016). This may be explained by plowing
producing a loose and rough soil surface. In the initial rainfall period, the soil particles were splashed
and rainfall water was stored in micro-depressions. Therefore, the initial infiltration rate was higher and
the runoff rate was lower, as shown by the results in Fig. 2 and Fig. 3. As the rainfall proceeded, the
surface was sealed by the impact of raindrops, so the infiltration rate decreased rapidly (Bissonnais,
1996; Shen et al., 2016). However, the infiltration rate remained at a higher level under WS compared
with TP in all of the rainfall simulation events under three slope gradients.

The cumulative infiltration amount showed that WS cover was an effective method for capturing

rainfall water. Table 1 shows that the cumulative infiltration amount was significantly higher under WS
than TP. The cumulative infiltration amounts under WS were 76.11, 75.95, and 75.83 mm with slopes of
5°, 10°, and 15°, respectively, i.e., 96.2%, 95.7%, and 94.8% rainfall infiltration, and thus there were
slight decreases as the slope gradient increased. A similar trend was also observed under TP, where the
cumulative infiltration amounts were 43.90, 32.53, and 28.14 mm with slopes of 5°, 10°, and 15°,
respectively, i.e., 53.7%, 40.8%, and 35.4% rainfall infiltration. According to these results, the
percentage of rainwater infiltrated indicated that WS cover significantly improved the capture of





rainwater. The standing stubble at the soil-atmosphere interface protected the soil surface from the direct
impact of raindrops and intercepted the rainfall near the soil surface, thereby give the rainwater more
time for infiltration. This might also explain the longer time required to generate runoff, the lower runoff
rates, and the higher infiltration amount under WS. Similar findings were obtained by Prosdocimi et al.
(2016) using barley straw mulch in Mediterranean vineyards, Jordán et al. (2010) using wheat straw
mulch under semiarid condition in southern Spain, Won et al. (2012) with straw mats covering soil in
laboratory rainfall simulations, Moreno-Ramón et al. (2014) in soil mulched with coffee husks in
laboratory rainfall simulations, and by Swella et al. (2015) using standing residues in a rainfed
agriculture system.
3.3 Soil loss
Table 1 shows the sediment concentration and total sediment loss in different treatment plots.
The sediment concentration under TP ranged between 8.14–14.90 g L$^{-1}$, which was significantly higher
than that under WS, i.e., 0.82–1.01 g L$^{-1}$. The total sediment loss varied in the same manner. The total
sediment loss under WS was 2.41–3.78 g m$^{-2}$, which was much lower than that under TP, i.e., 304.31–
731.23 g m$^{-2}$, and the sediment loss increased with the slope gradient. These results are consistent with
those expected because the stubble cover decreased soil losses (Hueso-González et al., 2015). Figure 4
also shows that the dynamic changes in the sediment concentration decreased dramatically under WS
compared with TP. Under TP, the sediment concentration increased rapidly in the first few minutes
because runoff was generated from the loose particles on the soil surface, before decreasing slightly. As
the rainfall simulation progressed, the sediment concentration reached a stable state because the loose
particles were exhausted. The observed changes in the sediment concentration during a rainfall event



followed a typical pattern, as reported by Jordán et al. (2010), Roth and Helming (1992), and Shen et al.
(2016). The differences in behavior between WS and TP may have been due to the standing stubble,
which absorbed the kinetic energy of raindrops and prevented their direct splashing (Gholami et al.,
2013). In addition, Jordán et al. (2010) noted that soil erosion was substantially reduced by high straw
residues due to decreased runoff detachment and an increased infiltration rate. Moreover, Won et al.
(2012) reported that substantial decreases in the sediment concentration under treatment with straw mat
cover was due to reduced runoff, thereby minimizing the overland flow transport capacity. Furthermore,
the wheat roots played an important role by reinforcing the soil and increasing infiltration (Katuwal et
al., 2013; Shinohara et al., 2016), which reduced the sediment concentration and sediment losses under
WS. Our results showed that TP disturbed the surface soil, which increased the available sediment
source and the likelihood of soil being detached and transported by raindrops and overland flow.
Ultimately, this may explain the higher sediment concentration under TP. These results agree with those
obtained by Engel et al. (2009), who found that tillage disturbed the soil surface, thereby facilitating the
transport of soil particles via overland flow. Celik (2005) also concluded that disturbance by tillage
could decrease the stability of soil particles so they were more likely to be detached. Vermang et al.
(2015) found that a rougher soil surface yielded a significantly higher sediment concentration because
the concentrated runoff could transport more particles.
3.4 Implications
The Loess Plateau is highly susceptible to soil erosion because of its erodible soil, sparse
vegetation cover, and concentrated summer precipitation. To address this problem, the Chinese
government launched the "Grain-for-Green" project in 1999 with the aim of converting steep sloping



(>25°) farmland into forest or grassland. However, 20% of the total farmland still has slopes of 15–25°
and 7% has slopes larger than 25° (NDRC et al., 2010). Our results confirm that WS delayed the runoff,
reduced soil erosion, and increased rainwater infiltration compared with TP. Figure 5 shows clearly that
WS was also effective in reducing the runoff rate and sediment concentration compared with TP. WS
significantly reduced the average runoff rate from 0.79 mm min$^{-1}$ (TP) to 0.08 mm min$^{-1}$ (WS), and the
average sediment concentration from 11.12 g L$^{-1}$ (TP) to 0.91 g L$^{-1}$ (WS). Therefore, the average
reductions in the runoff volume and sediment loss under WS were 92.27% and 99.39%, respectively,
compared with TP. Therefore, these results indicate that retaining stubble at the soil-atmosphere
interface can control soil erosion from farmland during the summer fallow period.
Furthermore, these experiments were conducted using small laboratory plots to test the
immediate effects of stubble on runoff and soil erosion under simulated rainfall conditions. It should be
noted that scale issues are important when interpreting results that affect practical applications (Sadeghi
et al., 2015). However, the results obtained in the present study can be used as comparative information
to indicate how WS may reduce the risk of soil erosion and increase rainwater capture on sloped
farmland. Mulumba and Lal (2008) noted that the WS cover varied according to the site-specific
conditions, so more studies are necessary to investigate a wider range of slopes, standing stubble height,
and different plot and field scales. Furthermore, it is necessary to determine the long-term impacts of
stubble cover on the soil properties (Bescansa et al., 2006; Lipiec et al., 2006), microbial activity (Song
et al., 2002), soil carbon content (Amundson et al., 2015; Lal, 1993; Van et al., 2007), and soil nutrient
cycling (Holland, 2004), as well as its relationships with soil erosion, crop productivity, and crop



biomass utilization. These studies may provide farmers and policymakers with a comprehensive
understanding of crop residue management under specific conditions in the Loess Plateau region.
**4. Conclusion**
This study investigated the effects of WS and TP on runoff and soil erosion in laboratory plots
under simulated rainfall with three different slope gradients. The results showed clearly that stubble can
be used as an effective management practice during the summer fallow period in the Loess Plateau
region. WS delayed the runoff initiation time, as well as decreased the runoff rate and runoff volume.
The sediment concentration and sediment losses were also decreased by WS compared with TP. In
addition, WS was highly beneficial because it absorbed the kinetic energy of raindrops and promoted
water infiltration. Thus, WS performed well according to all the variables considered in this study. The
reductions in runoff and the sediment concentration under WS were 91.0–93.2% and 99.2–99.6%,
respectively, compared with TP, and infiltration was 1.69–2.45 times higher under WS than TP. In
conclusion, WS was beneficial for reducing runoff and sediment losses, as well as increasing infiltration
under simulated rainfall. Stubble cover can be adopted as a management practice by farmers to control
soil erosion and promote rainwater conservation in the summer fallow period. Future studies should
evaluate the performance of this crop biomass by-product in diverse field conditions.

**Acknowledgement**
This study was financially supported by the National Natural Science Foundation of China (41271288).
The authors gratefully thank Dr Duncan E. Jackson for editing and improving the manuscript.

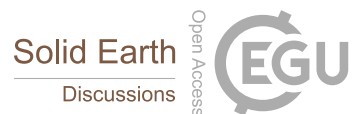

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



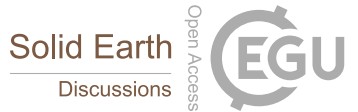

## Tables

Table 1 Selected chemical and physical properties of the soil

| Soil Type | Particle Size % | | | Soil Texture | Organic Matter % | pH | Wet Aggregate stability mm | CEC (Cation Exchange Capacity) cmol kg$^{-1}$ | CaCO$_3$ g kg$^{-1}$ |
|---|---|---|---|---|---|---|---|---|---|
| | Sand | Silt | Clay | | | | | | |
| Lou soil | 30.0 | 43.7 | 26.3 | clay loam | 1.33 | 8.2 | 1.4 | 18.1 | 74.6 |

Table 2 The time to runoff initiation, runoff rate, runoff volume, sediment concentration and loss, and

cumulative infiltration amount for WS and TP under each condition

| Slope | treatment | Time to runoff (min) | Runoff rate (mm min$^{-1}$) | Runoff volume (mm) | Sediment concentration (g L$^{-1}$) | Sediment loss (g m$^{-2}$) | Cumulative infiltration amount(mm) |
|---|---|---|---|---|---|---|---|
| 5 | WS | 23.68±4.29a | 0.09±0.00b | 3.00±0.51b | 0.82±0.19b | 2.41±0.14b | 76.12±0.74a |
| | TP | 4.78±0.36b | 0.66±0.01a | 37.86±0.67a | 8.18±0.04a | 304.31±5.93a | 43.90±1.11b |
| 10 | WS | 11.26±2.35a | 0.07±0.00b | 3.39±0.0.15b | 0.89±0.17b | 3.04±0.49b | 75.95±0.54a |
| | TP | 3.21±0.44b | 0.83±0.03a | 47.28±1.46a | 10.29±0.91a | 484.15±5.52a | 32.53±2.01b |
| 15 | WS | 7.07±2.78a | 0.08±0.00b | 4.16±0.22b | 1.01±0.37b | 3.78±0.76b | 75.83±2.86a |
| | TP | 2.99±0.09a | 0.90±0.01a | 51.42±0.69a | 14.90±1.26a | 731.23±11.66a | 28.14±0.59b |

For each slope gradient, the value is expressed as the mean ± the standard deviation, where the same lowercase

letters indicate no significant difference.

## Figures



Figure 1 Experimental runoff pot (a) and schematic of the rainfall simulator (b).

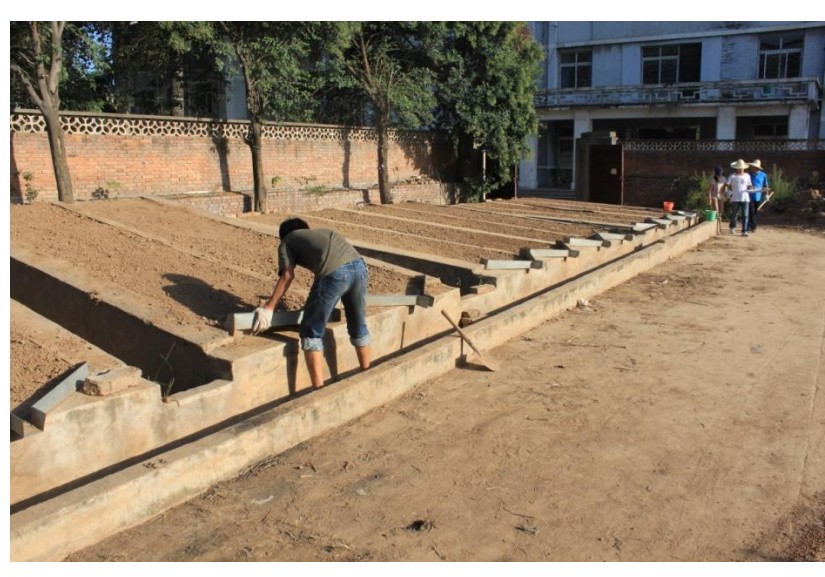


(a)

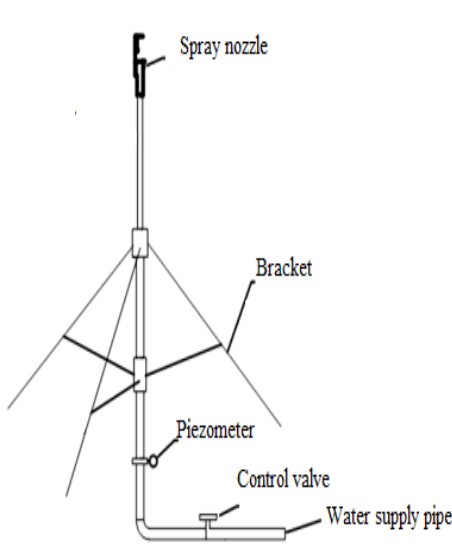


(b)




Figure 2 Dynamics of the runoff rate in WS and TP plot during the simulated rainfall event .







Figure 3 Dynamics of the infiltration rate in WS and TP plot during the simulated rainfall event.





Figure 4 Dynamics of the Sc in WS and TP plot during the simulated rainfall event.



Figure 5 Runoff rates versus sediment concentrations for WS and TP management treatment.

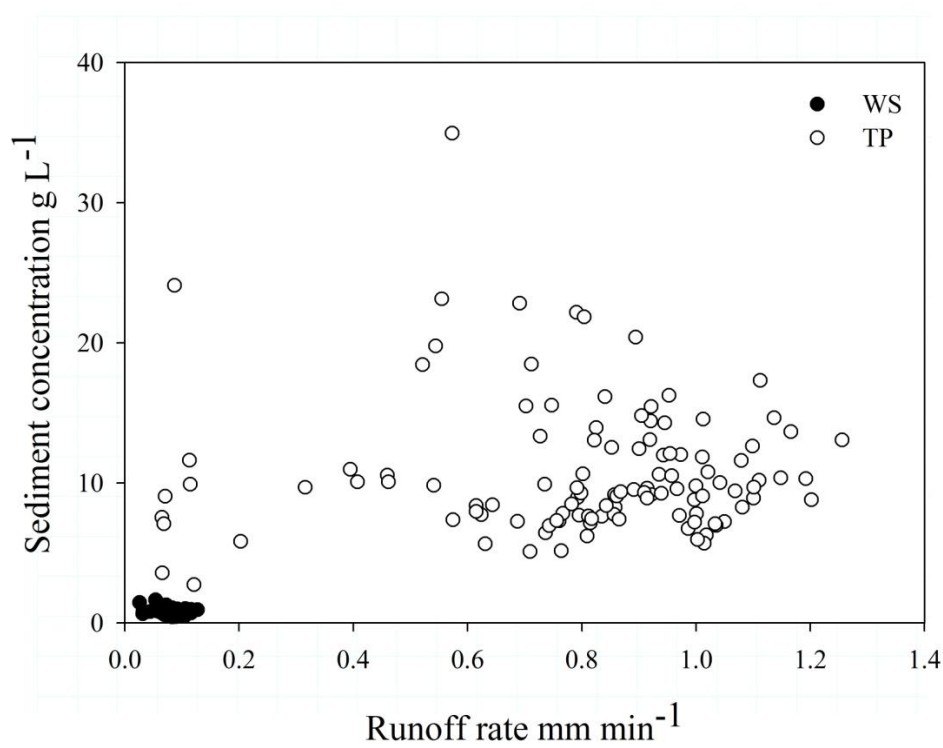
