# Peer review of "Published: 7 December 2016"

_Solid Earth, 2016_

## Referee Comment (RC1) · Anonymous Referee #1 · 30 Dec 2016

1. Line31-32: Please append some new studies in this field. 2. Line216: The infiltration amounts under WS were very similar, which show no significant among different slopes. 3. In the sector 3.3, I suggest authors to provide some insights about the performance of wheat stubble on soil loss at various slopes. The results also should introduce in the abstract section. 4. Line 269ãĂ̧271ïijŽThe language should further polish.

---

## Short Comment (SC1) · 30 Dec 2016

Soil erosion from sloped farmland is a serious environmental issue and a major sediment source of the Yellow River in the Loess Plateau region. Therefore, determining how the wheat stubble residues to control soil erosion during summer fallow period may be helpful to conserve soil and water. From this point of view, the manuscript was conducted on a meaningful research. My suggestions as fallowing: 1. Line 27 & Line147, "sloping farmland" change to "sloped farmland" 2. Line 257-259. "Vermang et al. (2015) found that. . . ." should be deleted. 3. Line 264. This sentence should add some references. 4. Line 459. In the figure title "runoff pot" should be "runoff plots". 5. In the figure 2, 3,4,and 5. I suggest use colors in these figures and the data symbols

should increase the size.

---

## Referee Comment (RC2) · Anonymous Referee #2 · 10 Jan 2017

[referee-annotated manuscript omitted]

---

## Author Comment (AC1) · 17 Jan 2017

Response to SE-2016-163-RC1 Thanks for your suggestions. We are appreciate for anonymous referee #1 comments concerning our manuscript entitled "Effects of wheat stubble on runoff, infiltration, and erosion of farmland in the Loess Plateau, China subjected to simulated rainfall" (ID: SE-2016-163). We have studied comments carefully and have made correction. The main corrections in the paper according to the reviewer's comments are as follows: 1. Line 31-32: Please append some new studies in this field. Response: Line 31-32: Approximately 60% of the total watershed sediment and runoff is derived from sloped farmland due to natural and human factors, such as the precipitation intensity, geomorphology, and soil management practices, which all

contribute to farmland degradation (Keesstra et al., 2016; Liu et al., 2012; Nishigaki et al., 2017; Zhao et al., 2016; Ziadat and Taimeh, 2013).

2. Line 216: the infiltration amounts under WS were very similar, which show no significant among different slopes. Response: Line222-224: The cumulative infiltration amounts under WS were 76.11(5°), 75.95(10°), and 75.83(15°) mm with no significant difference among different slopes and 96.2%, 95.7%, and 94.8% rainfall infiltrated into soil respectively.

3. In the sector 3.3. I suggest authors to provide some insights about the performance of wheat stubble on soil loss at various slopes. The results also should be introduce in the abstract section. Response: Line 237-247: Table 1 shows the sediment concentration and total sediment loss in different treatment plots. The sediment concentration under TP ranged between 8.18 (5°) to 14.90 g L–1(15°), which was significantly higher than that under WS, i.e., 0.82(5°)-1.01 g L–1(15°). The total sediment loss varied in the same manner. The total sediment loss under WS was 2.41(5°)-3.78 g m–2(15°), which was much lower than that under TP, i.e., 304.31(5°)-731.23 g m–2(15°). The sediment loss in WS and TP was increased by 56.8%, 140.3%, respectively, as the slope gradient increased from 5° to 15°. This indicated that wheat stubble has greater effectiveness in reducing sediment loss at higher slope gradient. In this study, the sediment control effect of wheat stubble was supported by the infiltration capacity, leading to a significantly reduced runoff in comparison to TP despite a higher slope gradient. These results are consistent with those expected because the stubble cover decreased soil losses (Hueso-González et al., 2015). We also corrected in the abstract section. Line 20-26: The sediment concentration was significantly lower with WS than TP. Compared with TP (304.31-731.23 g m–2), the sediment losses were reduced dramatically in WS (2.41-3.78 g m–2) and the sediment loss slightly increased with slope, however, it was greatly increased as slope increased in TP. These results revealed that the stubble cover was the main factor reducing runoff and sediment losses and improving infiltration, and that stubble showed a great potential to control erosion and conserve

soil and water resources during the summer fallow period in Loess Plateau region.

4. Line269-271: the language should further polish. Response: Line 275-278: WS significantly reduced the average runoff volume from 45.52 mm (TP) to 3.52 mm (WS), and the average sediment loss from 506.56 g m–2 (TP) to 3.08 g m–2 (WS). Therefore, the runoff volume and sediment loss was reduced by 92.27% and 99.39% by the wheat stubble respectively.

Please also note the supplement to this comment:
http://www.solid-earth-discuss.net/se-2016-163/se-2016-163-AC1-supplement.pdf

---

## Author Comment (AC2) · 17 Jan 2017

Thanks for your suggestions. We are appreciate for the comments concerning our manuscript entitled "Effects of wheat stubble on runoff, infiltration, and erosion of farmland in the Loess Plateau, China subjected to simulated rainfall" (ID: SE-2016-163). We have studied comments carefully and have made correction. The main corrections in the paper according to the reviewer's comments are as follows:

1. Line 27 and Line 147: "sloping farmland" change to "sloped farmland" Repsonse: Line 30: Soil and water losses from agricultural land, particularly sloped farmland, are regarded as major environmental threats to ecosystem sustainability on the Loess Plateau, China. Line 153-154: TP is typically used by local farmers on the sloped

farmland in the Loess Plateau region.

2. Line 257-259: ""Vermang et al. (2015) found that…" should be deleted. Response: this sentence has been deleted in the manuscript.

3. Line 264 this sentence should add some references. Response: To address this problem, the Chinese government launched the "Grain-for-Green" project in 1999 with the aim of converting steep sloping (>25°) farmland into forest or grassland (Cao et al., 2009; Wang, 2015).

4. Line459 In the figure title "runoff pots" should be "runoff plots" Response: Line 479: Figure 1 Experimental runoff plots (a) and schematic of the rainfall simulator (b).

5. In the figure2-5, I suggest use colors in these figures and the data symbols should increase the size: Response: the figures have been reset according to the suggestions.

Please also note the supplement to this comment:
http://www.solid-earth-discuss.net/se-2016-163/se-2016-163-AC2-supplement.pdf

1    Figure 2 Dynamics of the runoff rate in WS and TP plot during the simulated rainfall event .

[Figure]

4

**Fig. 1.**

---

## Author Comment (AC3) · 17 Jan 2017

Thanks for your suggestions. We are appreciate for anonymous referee #2 comments concerning our manuscript entitled "Effects of wheat stubble on runoff, infiltration, and erosion of farmland in the Loess Plateau, China subjected to simulated rainfall" (ID: SE-2016-163). We have studied comments carefully and have made correction. The main corrections in the paper according to the reviewer's comments are as follows:

The paper is fine and just need some improvements for the introduction to make more accessible for the readers of LDD.

Response: The introduction has been improved and added some LDD references as

follows:

Line30-35: Soil and water losses from agricultural land, particularly sloped farmland, are regarded as major environmental threats to ecosystem sustainability on the Loess Plateau, China. Approximately 60% of the total watershed sediment and runoff is derived from sloped farmland due to natural and human factors, such as the precipitation intensity, geomorphology, and soil management practices, which all contribute to farmland degradation (Keesstra et al., 2016; Liu et al., 2012; Nishigaki et al., 2017; Zhao et al., 2016; Ziadat and Taimeh, 2013).

Line 55-69: To reduce soil erosion from farmland, numerous studies have considered the roles of surface cover and soil management practices in conservation agriculture worldwide (Bescansa et al., 2006; Jordán et al., 2010; Prosdocimi et al., 2016; Swella et al., 2015; Won et al., 2012). Retaining a surface covered with a layer of crop residues is a suitable management practice for preventing soil losses and conserving rainwater on farmland. Thus, Gholami et al. (2013) and Kukal and Sarkar (2010) investigated the effects of straw mulch on soil erosion under laboratory simulated rainfall conditions, and concluded that straw mulch was effective in delaying the runoff initiation time, as well as reducing splash erosion, runoff and soil losses. Nishigaki et al. (2016) investigated the effects of vegetative residues on runoff and soil losses under field experiment condition, the results showed that surface mulch reduced soil losses caused by raindrop detachment and also suppressed runoff generation. Mwango et al. (2016) studied the effectiveness of mulching on soil erosion and nutrient losses, and found that mulches had greater potential in decreasing runoff, soil and nutrient losses and similar results were also observed by Wang et al. (2016) in the Jujube plot. Jordán et al. (2010) showed that the application of wheat straw mulch could improve the physical and chemical properties of soil, as well as reduce runoff and soil losses in cultivated land.